# Dr ChatGPT tell me what I want to hear:
## How different prompts impact health answer correctness

**Bevan Koopman**
CSIRO and The University of Queensland
Brisbane, Australia
bevan.koopman@csiro.au

**Guido Zuccon**
The University of Queensland
Brisbane, Australia
g.zuccon@uq.edu.au

## Abstract

This paper investigates the significant impact different prompts have on the behaviour of ChatGPT when used for health information seeking. As people more and more depend on generative large language models (LLMs) like ChatGPT, it is critical to understand model behaviour under different conditions, especially for domains where incorrect answers can have serious consequences such as health. Using the TREC Misinformation dataset, we empirically evaluate ChatGPT to show not just its effectiveness but reveal that knowledge passed in the prompt can bias the model to the detriment of answer correctness. We show this occurs both for retrieve-then-generate pipelines and based on how a user phrases their question as well as the question type. This work has important implications for the development of more robust and transparent question-answering systems based on generative large language models. Prompts, raw result files and manual analysis are made publicly available at https://github.com/ielab/drchatgpt-health_prompting.

## 1 Introduction

Prompt-based generative large language models, such as ChatGPT, can be used to answer complex natural language questions, often with impressive effectiveness. Where previously, a search engine was the main tool available, now users can choose to use a large language model (LLM). While queries to search engines are typically short and ad-hoc, prompts to a LLM can be far longer and richer. This can be a double edged sword: on the one hand, far more information is available to the model to generate a good answer; on the other hand, LLMs suffer from hallucinations (Alkaissi and Mcfarlane, 2023) and incorrect, biased or misleading prompt information (e.g., obtained through a retrieved-then-generate pipeline) may derail the model's ability to give a good answer. While in

some domains this may not matter, in this paper we investigate a domain where this could have serious consequences — health. In particular, we consider the case of a health consumer (i.e., non professional) using ChatGPT to understand whether a treatment $X$ has a positive effect on condition $Y$.

Our evaluation is done using the TREC Misinformation track, a resource that contains complex health questions, often pertaining to misinformation regarding the efficacy of treatments, and that are typical of a subset of information seeking tasks undertaken by health consumers. Even though the risks of searching for health information online are well documented (Benigeri and Pluye, 2003; White and Horvitz, 2009; Zuccon et al., 2015), users still continue to seek health information through online mediums (Fox, 2011). Research on the effectiveness of using traditional search engines for health answers does exist (Zuccon et al., 2015), but the parallel for LLMs does not. This paper aims at investigating this by considering two main experimental conditions and one specific, popular LLM, ChatGPT, currently being relied upon by end-users:

**Question-only:** ChatGPT is asked to provide an answer to a health question without further information provided to the model; and

**Evidence-biased:** ChatGPT is asked to provide an answer to a health question after being provided a prompt with information from a web search result (document) containing information about treatment $X$ and condition $Y$. This provides insights into a typical retrieve-then-generate setup, whereby a search engine provides results in a prompt to the LLM.

We investigate cases where the document provided in the prompt either supports the use of the treatment (supporting evidence) or where the document dissuades from the use of the treatment (contrary evidence). Given these settings, we can then evaluate our two main research questions:

**RQ1 - Question-Only Effectiveness:** How effective is ChatGPT at answering complex health information questions typical health consumers would ask? How does question phrasing impact this effectiveness?

**RQ2 - Evidence-Biased Effectiveness:** How does biasing ChatGPT by prompting with supporting and contrary evidence influence answer correctness?

The paper is divided in two parts around these two questions. For question-only effectiveness (RQ1), we show that ChatGPT is actually quite effective at answering health related questions (accuracy 80%), but this effectiveness hides a systematic bias based on both how the question is phrased and whether the correct answer to the question is a "Yes" or "No" that majorly degrades effectiveness.

When biasing the prompt with supporting or contrary evidence (RQ2), we find that: (1) the evidence often is capable of overturning the model answer about a treatment, showing prompts have real impact on answer correctness; and (2) both supporting and contrary evidence can have a detrimental effect on answer correctness, reducing accuracy to 63%.

Previous work has shown that engineering prompts has an important impact on the effectiveness of LLMs like ChatGPT (Liu et al., 2023). This paper adds to that understanding by contributing that it is not just the "form" of the prompt that matters (e.g., the clarity of the instructions contained in the prompt, or the stance expressed in the prompt), but also that the correctness of information contained in the prompt can highly influence the quality of the output of the LLMs. This is important when LLMs are integrated in a retrieve-then-generate[1] pipeline (Chen et al., 2017; Karpukhin et al., 2020; Lewis et al., 2020), where information related to the question is first identified from a corpus (e.g., the Web), and this is then passed to the model via the prompt to inform the model's output.

## 2 Related Work

### 2.1 Health information seeking on the Web.

People rely on information on the web to make health decisions (Fox, 2011). Researching how they seek health information and advice on the web,

---

[1]Also referred to as read-then-generate or retrieval-enhanced generation.

including using search engines and symptom checkers, has therefore attracted a substantial amount of research (Toms and Latter, 2007; White and Horvitz, 2009; Miller and Pole, 2010; White, 2013; Stanton et al., 2014; Lopes and Ribeiro, 2015; Semigran et al., 2015; Pogacar et al., 2017). The effectiveness of these technologies has also been monitored and assessed through several studies (Semigran et al., 2015; Zuccon et al., 2015; Jimmy et al., 2018; Cross et al., 2021), and new technologies have also been deployed; e.g., the health information cards now commonly displayed by major commercial search engines (Gabrilovich, 2016; Jimmy et al., 2019b,a). Health chatbots based on pre-LLMs technology have also been proposed and investigated (You et al., 2023); however, extensive evaluation and understanding of chatbot solutions that rely on current state-of-the-art methods in conversational generative LLMs is lacking.

### 2.2 ChatGPT for health question-answering.

In this paper, we empirically study the effectiveness of ChatGPT in answering consumer health questions. Limited prior work looked at how well ChatGPT performs on medical exam questions (Gilson et al., 2022; Kung et al., 2023). Instead, we consider questions the general public may ask, which are quite different to medical exams. In addition, these previous studies did not consider the impact of prompting with evidence (our RQ2).

Gilson et al. (2022) presented a preliminary evaluation of ChatGPT on questions from the United States Medical Licensing Examination (USMLE) Step 1 and Step 2 exams. The prompt only included the exam question, and ChatGPT answer was evaluated in terms of accuracy of the answer, along with other aspects related to logical justification of the answer and presence of information internal and external to the question. ChatGPT effectiveness was comparable to a $3^{rd}$ year medical student.

Nov et al. (2023) compared ChatGPT responses to those supplied by a healthcare provider in 10 typical patient-provider interactions. 392 laypeople were asked to determine if the response they received was generated by either ChatGPT or a healthcare provider. The study found that it was difficult for participants to distinguish between the two, and that they were comfortable using chatbots to address less serious health concerns — supporting the realistic setting of our study.

Benoit (2023) evaluated ChatGPT in the diagnosis and triage of medical cases presented as vignettes[2] and found ChatGPT displayed a diagnostic accuracy of 75.6% and a triage accuracy of 57.8%.

De Angelis et al. (2023) recognised that ChatGPT can be used to spread misinformation within public health topics. In our study, we empirically demonstrate this to be the case, including the systematic biases present in the current ChatGPT model that exacerbate this risk. In addition, we show how misinformation can be injected into ChatGPT's prompt and this definitively impacts answer correctness, with potentially dire consequences.

## 2.3 Prompt construction for LLMs.

One of the aims of our study is to evaluate the impact of ChatGPT answer correctness with respect to different prompt strategies. The effect of prompting LLMs is attracting increasing attention, especially with respect to the so called practice of "prompt engineering"; i.e., finding an appropriate prompt that makes the language model solve a target downstream task. Prompt engineering, as opposed to fine-tuning, does not modify the pretrained model's weights when performing a downstream task (Liu et al., 2023). Prompt engineering is commonly used to enable language models to execute few-shot or zero-shot learning tasks, reducing the need to fine-tune models and rely on supervised labels. Furthermore, a "prompt-learning" approach can be applied as follows: during inference, the input $x$ is altered using a template to create a textual prompt $x'$, which is then provided as input to the language model to generate the output string $y$. The typical prompt-learning setup involves constructing prompts with unfilled slots that the language model fills to obtain a final string $\hat{x}$, which is then used to produce the final output $y$ (Liu et al., 2023).

In our study, we do not perform prompt-learning: we instead use the prompt to pass external knowledge to the model and measure how this changes the answers it generates. Related to this direction, but in the context of few-shot learning, Zhao et al. (2021) observed that the use of prompts containing training examples for a LLM (in particular GPT-3) provides unstable effectiveness, with the choice of prompt format, training examples and their order being major contributors to variability in effective-

[2]A clinical vignette is a brief patient summary that includes relevant history, physical exam results, investigative data, and treatment.

```
[question_text]
Answer <Yes>, <No>, and provide an
explanation afterwards.
```

Figure 1: GPTChat prompt format for determining general effectiveness (RQ1) on TREC Misinformation topics.

ness. We have similar observations in that in RQ2 we vary the documents provided as evidence in the prompt and find that two documents can have widely different effects on the answer of the model despite having the same stance about the topic of the question.

## 3 RQ1 - Question-only Effectiveness

Our first research question relates to how effective ChatGPT is in answering complex health information questions. Measuring this serves two purposes: (i) it answers the basic question of how effective is this LLM in health information seeking tasks; and (ii) it provides the baseline effectiveness of the model when relying solely on the question; i.e., without observing any additional knowledge in the prompt. We also consider ChatGPT effectiveness on Yes vs No type questions and how question phrasing can impact ChatGPT.

### 3.1 Methods

We use 100 topics from the TREC 2021 and 2022 Health Misinformation track (Clarke et al., 2021) with associated ground truth. Each topic relates to the efficacy of a treatment for a specific health issue. The topic includes a natural language question, e.g., "Does apple cider vinegar work to treat ear infections?", and a ground truth answer, either 'Yes' or 'No'. Ground truth answers were assigned based on current medical evidence.

We issue each question to ChatGPT as part of the prompt shown in Figure 1, which instructs the model to provide a Yes/No answer and associated explanation. We then evaluate the correctness of the answer by comparing the ChatGPT answer with the TREC Misinformation Track ground truth.

In addition to asking for a Yes/No answer, we also run another experiment instructing the model to provide a Yes/No/Unsure option to determine the impact of adding the unsure option. For this, the prompt in Figure 1 was modified to add the Unsure option. Since TREC Health Misinformation only provides Yes/No answers, Unsure is treated

as an incorrect answer in our evaluation. The Unsure option was investigated to understand how often ChatGPT responded with Unsure if given the option. Furthermore, our results will reveal that Unsure answers could potentially lead users to adopt false health claims with serious consequences. For this reason as well, we treat Unsure as incorrect.

Processing the response from ChatGPT was done as follows: If the ChatGPT response started with the terms "Yes", "No", or "Unsure" then these were recorded automatically. (Most cases fit this condition.) For the few cases where the response did not start with the terms "Yes", "No" or "Unsure", we manually assessed these responses. The manual assessment was done by two annotators (authors). While there were no disagreements, annotators flagged some cases for joint discussion and adjudication.

Questions in the TREC Misinformation dataset are in the form "Can $X$ treat $Y$?". Our initial results, discussed below, revealed a systematic bias in ChatGPT behaviour dependent on whether the ground truth was a Yes or No answer. To further investigate this effect we conducted an additional experiment whereby we manually rephrased each question to its reversed form: "Can $X$ treat $Y$?" becomes "$X$ can't treat $Y$?". At the same time we inverted the corresponding ground truth to match. Note that this was just a rephrasing from the affirmative to the negative, without changing the contents. The hypothesis here was that the phrasing of the question to match either a Yes or No answer has a significant impact on ChatGPT behaviour and effectiveness.

To control for randomness in ChatGPT responses, we repeated the above experiments 10 times and averaged the results (see the Limitations section for further discussion).

### 3.2 Results

The effectiveness of ChatGPT is shown in Figure 2. Overall accuracy was 80%. ChatGPT answered "Yes" and "No" a similar number of times (55% vs 45%) and its error rate was similar between the two. When allowing for an "Unsure" option, accuracy was much lower at 33% and the model hardly ever gave a "No" answer. This lower accuracy is also a by-product of our evaluation which treated "Unsure" as an incorrect answer when comparing with the TREC Health Misinformation ground truth. We felt this was appropriate as when examining "Un-

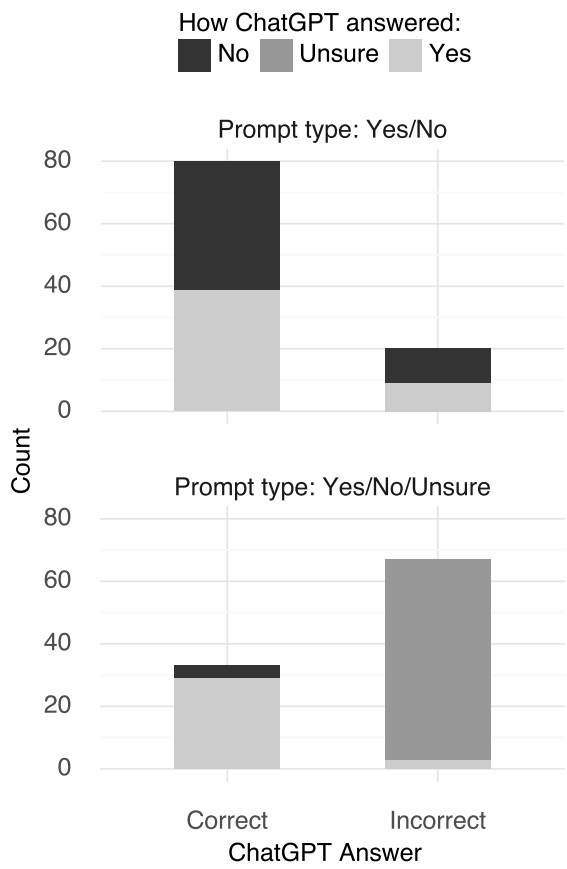

| Prompt | Overall Accuracy |
|---|---|
| Yes/No | 0.80 |
| Yes/No/Unsure | 0.33 |

Figure 2: Effectiveness of ChatGPT when prompting for "Yes/No" and "Yes/No/Unsure" answers to TREC Misinformation questions.

sure" answers from a qualitative standpoint, we found that while ChatGPT answered Unsure, the explanation given was not neutral. Consider the question "Will drinking vinegar dissolve a stuck fish bone?", which has a ground truth answer of No. The ChatGPT response was "Unsure. Drinking vinegar may help break down the fish bone, but it's not a guaranteed solution.". This was clearly not a definitive No and could well drive a user to take an unfounded and potentially dangerous treatment (Shields et al., 2016). Further qualitative assessment revealed that other answers were specifically formulated to cover the whole spectrum of opinions on the topic. Covering all opinions would potentially fuel anchoring bias (Azzopardi, 2021; Lau and Coiera, 2007; White, 2013); i.e., the

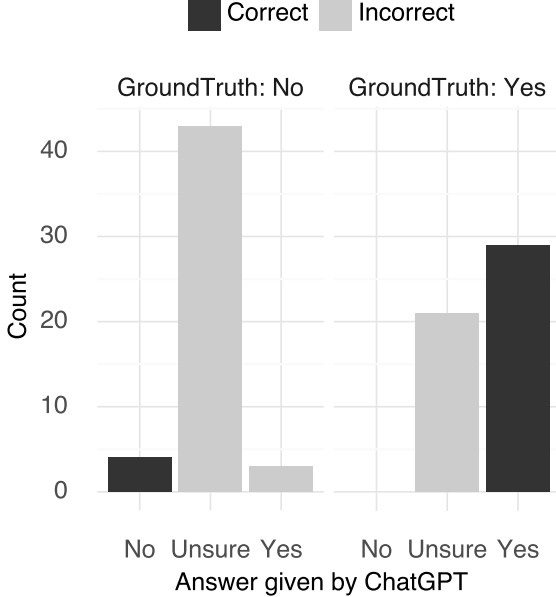

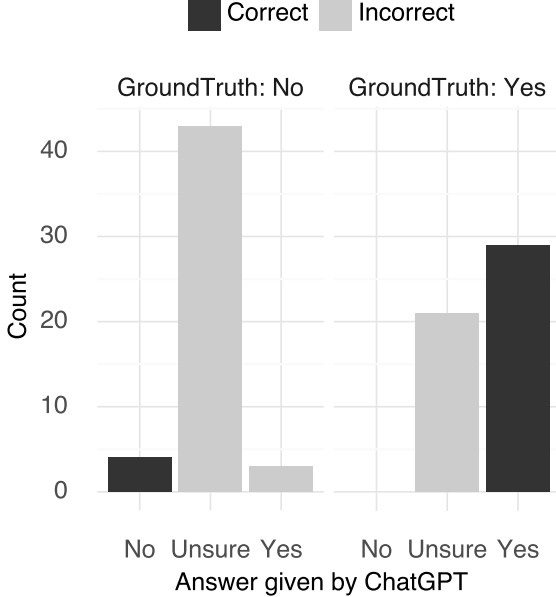

Figure 3: If the correct answer to a health question is Yes, ChatGPT is much more likely to get it correct. ChatGPT is much more likely to respond with Unsure to questions where the correct answer is No.

user perceives the answer as affirming their preconceived answer/stance. Anchoring bias is shown to be a major challenge when supporting consumers seeking health information (White, 2013).

Besides the decision to treat Unsure as Incorrect, we did not perform an in depth analysis of these explanations (e.g., to verify whether the claims made in the explanation were true or were hallucinations (Ji et al., 2022; Bang et al., 2023)), because we lack the required medical expertise. We plan to investigate this aspect in future work through engagement with medical experts. A brief analysis of these explanations, however, revealed that they often contain remarks about the presence of limited scientific evidence (or even conflicting evidence) with respect to a treatment option for a condition. The answers also often contain a suggestion to contact a health practitioner for further review of the advice.

TREC Misinformation ground truth was either Yes or No in answer to the question. On analysis of ChatGPT answers for the Yes/No/Unsure prompt, we discovered a significant difference in ChatGPT behaviour based on whether the ground truth answer was a Yes or No. Figure 3 divides the results according to whether the ground truth was Yes or No. For Yes answers (right subplot), ChatGPT was

far more effective, answering Yes correctly most often, then answering Unsure, but never answering No. In contrast, for No answers (left subplot), ChatGPT was far less effective and answered Unsure the vast majority of the time. This behaviour represents a systematic bias in ChatGPT on whether the answer to a user's question is a Yes or a No.

There is an important difference to a user on getting a Yes or No answer correct. Recall that TREC Misinformation most often contained questions of the form "Can $X$ treat $Y$?". If the ground truth is a Yes, ChatGPT will say Yes or Unsure (it never answered No to Yes questions). The resulting user could seek the treatment $X$ or remain unsure. Either way they will not be following an unfounded medical claim. Now consider the opposite case where the ground truth is a No. Such questions often pertain to false health claims (e.g., "Will drinking vinegar dissolve a stuck fish bone?"). For such questions, ChatGPT is highly likely to answer Unsure. From qualitative assessment of ChatGPT's explanation, we know that Unsure answers often contain some rationale for a Yes and some for a No; a user with any preconceived bias on the topic may adopt their viewpoint. If they adopt the Yes viewpoint then they may well seek the treatment for a false medical claim (e.g., electing to drink the vinegar to dissolve the stuck fish bone). Seeking treatments based on false health claims can have adverse health effects (indeed, ingestion of vinegar can in fact be fatal (Shields et al., 2016)).

Having discovered a systematic difference in ChatGPT behaviour for Yes or No questions, we turn to results for manually reversing the phrasing of questions and ground truth (e.g., "Can $X$ treat $Y$?" becomes "$X$ can't treat $Y$?"). Figure 4 shows the effectiveness of ChatGPT on answering the reversed queries. (This figure can be compared to Figure 2, which was obtained for the original questions). Overall, ChatGPT was far less effective for the manually reversed questions compared to the original phrasing. Accuracy dropped from 0.80 to 0.56 for the Yes/No prompt and 0.33 to 0.04 for the Yes/No/Unsure prompt. We observe from Figure 4 that ChatGPT very seldom answered Yes at all to any of the questions. This is in contrast to the results obtained for the original questions (Figure 2) where Yes answers were quite common. Recall that we simply rephrased the question from an affirmative to a negative (as well as switching the associated ground truth). While the core content

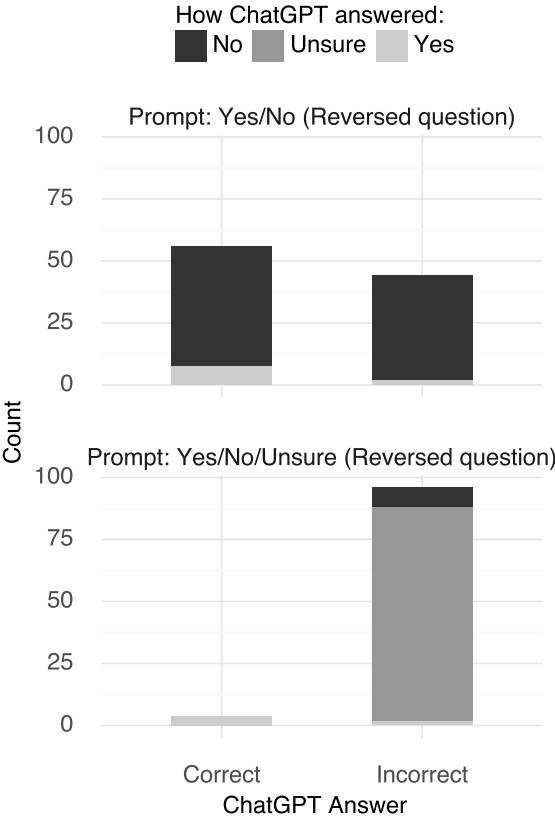

| Prompt | Overall Accuracy |
|---|---|
| Yes/No | 0.56 |
| Yes/No/Unsure | 0.04 |

Figure 4: Effectiveness of ChatGPT when prompting after rephrasing TREC Misinformation questions to reversed form (e.g., "Can $X$ treat $Y$?" becomes "$X$ can't treat $Y$?"). When comparing to the original questions from Figure 1 we note that simply rephrasing the question strongly degraded answer correctness.

of the question was unchanged, the rephrasing had a huge impact on ChatGPT responses. Previous work has shown that there is considerable variation in how people pose health questions to search engines (Zuccon et al., 2015; Koopman et al., 2017). This variation leads to significant impact in search engine effectiveness. Our results suggest this is also the case for ChatGPT. However, for ChatGPT we find that this effect is quite insidious in that it arises from how the question is phrased rather than the actual keywords authored by the user.

## 4 RQ2 - Evidence-biased Effectiveness

Our second research question investigates the impact on answer correctness of biasing ChatGPT by prompting with (i) supporting evidence; and (ii) contrary evidence. Measuring this allows us to determine the impact of providing external information via the prompt. It helps also to understand how ChatGPT might perform in a retrieve-then-generate pipeline.

### 4.1 Methods

Supporting and contrary evidence was taken as individual documents from the TREC Misinformation track ground truth (qrels). Documents judged as 2 ("Supportive") were selected as supporting evidence. Documents judged as 0 ("Dissuades") were taken as contrary. The process to issue the evidence biased prompt to ChatGPT was as follows:

1. For each TREC topic, we selected a maximum of 3 supportive and 3 contrary evidence documents from the TREC qrels;[3]

2. For each document, we generated the prompt in Figure 5, including both the question and the evidence text in quotes;

3. Although the prompt explicitly asked for a Yes/No answer, we found that in many cases ChatGPT did not use these words, but did provide the answer in another way (e.g., "Inhaling steam can help alleviate the symptoms of a common cold"). We manually mapped these responses to Yes, No, or Unsure. All answers were assessed by two of the authors and discussion took place to resolve disagreement.

4. Once ChatGPT provided an answer, we then evaluated the correctness of the answer by comparing to the TREC Misinformation Track ground truth.

We again added a Yes/No/Unsure prompting option, similar to RQ1. We used the 35 topics from TREC 2021 Health Misinformation track that contained document-level relevance assessments.[4]

---

[3]We took the first three documents according to the official TREC qrel file. All topics had at least 3 supporting evidence passages. 23 topics had at least 3 contrary evidence passages. 2 topics had only 2 contrary evidence passages. 6 topics had only 1 contrary evidence passage. 4 topics had 0 contrary evidence passages.

[4]Document-level qrels are not available for TREC 2022 at the time of writing.

```
[question_text]

A web search for this question has returned the
following evidence, which I provide to you in
quotes:

"[document_text]"

You MUST answer to my question with one of the
following options ONLY: <Yes>, <No>. Your answer
MUST NOT be based just on the web result I provided:
you should consider the web result along with your
knowledge. Please also provide an explanation for
your answer.
```

Figure 5: ChatGPT Prompt used to determine what impact a supportive or contrary document has on answer correctness.

The documents were web pages from the no-clean version of the C4 dataset.[5] Some documents exceeded the maximum token limit of ChatGPT and had to be trimmed. We used the NLTK `word_tokenize` method to count the number of tokens. Then, through experimentation, we identified that a limit of 2,200 NLTK tokens was about the maximum we could use to concatenate with the remainder of the prompt and issue to ChatGPT without encountering problems with the input size limit. An analysis of document length showed that the median was 1,845 tokens, and trimming only occurred on three occasions.

Figure 9 of the appendix contains two actual prompts we used, one with support and one with contrary evidence. Experiments were repeated 10 times and results averaged to control from randomness in ChatGPT.

### 4.2 Results

The effectiveness of ChatGPT with evidence-biased prompting is shown in Figure 6. Overall accuracy was 63% — far less than the 80% of the simpler, question-only prompt. In addition, when evidence-biasing in the prompt, ChatGPT was much more likely to give a "Yes" answer (62% Yes vs 55% Yes in RQ1). If the prompt allowed for an "Unsure" answer then ChatGPT answered Unsure 62% of the time, reducing accuracy to 28%.

Figure 7 shows a breakdown of effectiveness after evidence-biasing. Specifically, how the answer changed compared to RQ1's question-only condition: "Flipped" indicates that ChatGPT's answered opposite (Yes to No or No to Yes) after evidence-

---
[5]C4 contains≈1B english extracts from the April 2019 snapshot of Common Crawl.

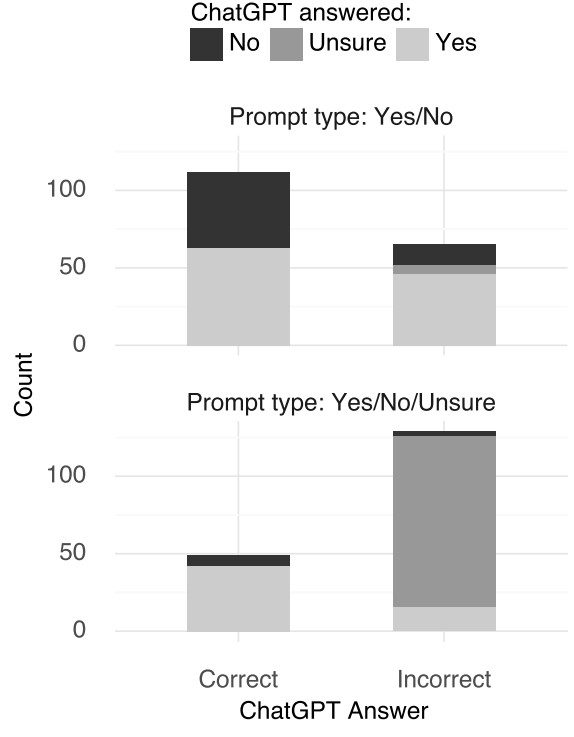

Figure 6: Effectiveness of ChatGPT when prompting with either a supporting or contrary evidence.

| Prompt | Overall Accuracy |
|---|---|
| Yes/No | 0.63 |
| Yes/No/Unsure | 0.28 |

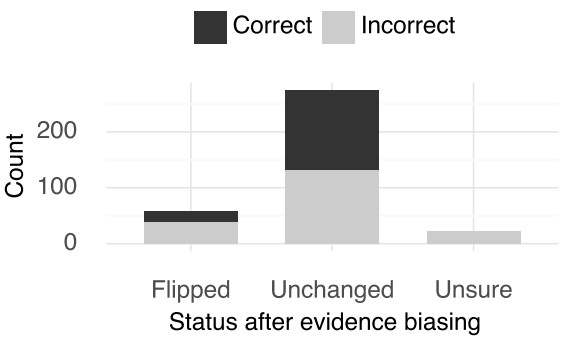

Figure 7: How evidence-biased prompting changed ChatGPT answers when compared with question-only prompting.

biasing; "Unchanged" means the answer matched RQ1's question-only condition; "Unsure" indicates where ChatGPT did not provide a Yes/No answer. We observe that when ChatGPT changed its answer (i.e., Flipped), it was wrong 68% of the time. In conclusion, evidence-biasing can flip the answer

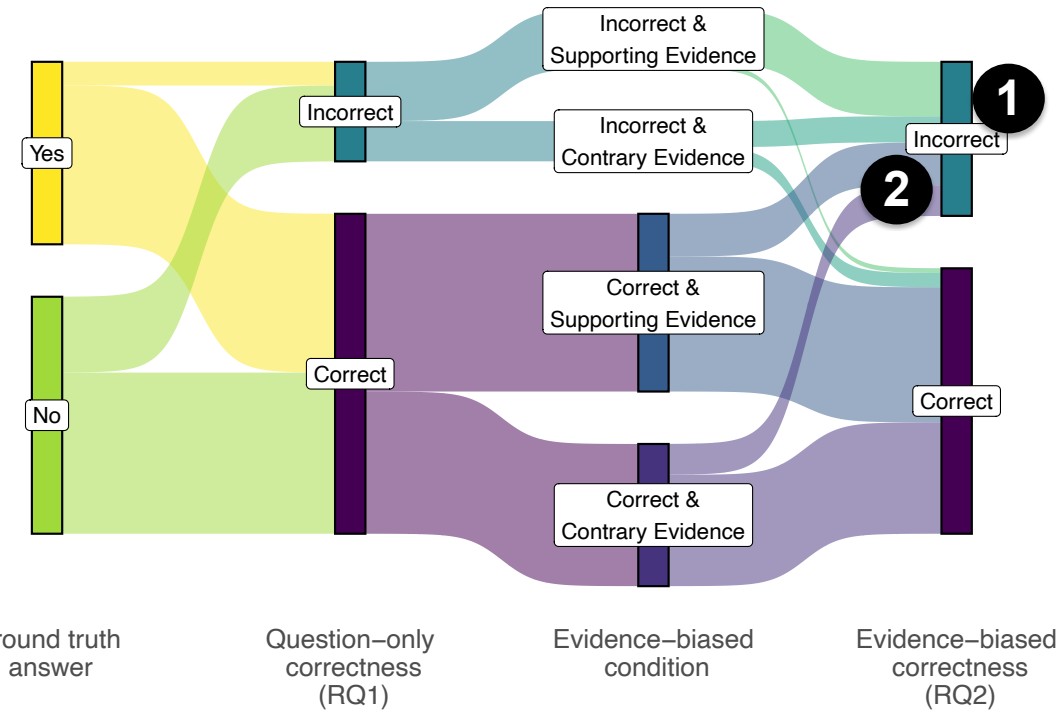

Figure 8: Sankey diagram showing the breakdown of all results. From the left, topics are divided by ground truth answer (Yes or No); next topics are divided according to RQ1 question prompting (Correct or Not Correct); next the prompt is evidence biased (Supporting and Contrary); finally, post evidence-biased breakdown is shown.

and it's typically for the worse.

Figure 8 provides a detailed analysis of the answer behaviour of ChatGPT with respect to the two conditions: question-only and evidence-biased. This analysis provides a number of notable insights. First, consider the incorrect answers provided in the evidence-biased condition (label ❶ in the Sankey diagram). These were cases where providing the question only in RQ1 was incorrect and providing evidence in RQ2 was also incorrect. In these cases, providing supporting evidence to try to overturn an incorrect answer did not flip it to a correct answer. In fact, providing the evidence (contrary or supportive) had little impact: nearly all the Incorrect cases from the RQ1 column flowed to the Incorrect column of RQ2.

Now consider the case where adding evidence caused the answer to flip from correct (RQ1) to incorrect (RQ2) (label ❷). This actually occurred more often when supporting evidence was provided, rather than contrary evidence, but the difference was minimal. The key takeaway here is that providing evidence degrades answer accuracy, even when that evidence is supporting.

In conclusion, prompting with evidence did not help flip incorrect answers to correct and actually caused correct answers to flip to incorrect. This may serve as a warning for relying on retrieve-

then-generate pipelines to reduce hallucinations and improve answer correctness.

## 5 Conclusions

We have examined the correctness of ChatGPT, a rapidly emerging generative large language model, in answering complex health information questions regarding the effectiveness of a treatment for a condition. We did this in two settings: when only the question is presented to ChatGPT (question-only) and when the question is presented along with evidence (evidence-biased); i.e., a web search result retrieved when searching for the question. Importantly, we controlled the stance used by the question, and whether the evidence was in supporting or contrary to the treatment. This in turn allowed us to understand the effect of question phrasing and of providing evidence in the prompt. ChatGPT answer accuracy was 80% when relying solely on the question being posed, and no additional evidence. However, (i) ChatGPT displays a systematic bias based on whether the question has a Yes or No answer, (ii) simply rephrasing the question to its negative form strongly degrades effectiveness, and (iii) providing evidence in the prompt degrades answer accuracy, even if the evidence is correct.

We specifically highlight the last result: injecting evidence into the prompt, akin to retrieve-then-

generate pipeline, can heavily influence the answer — more importantly, it reduced ChatGPT accuracy to only 63%. This finding is at odds with the idea of relying on retrieve-then-generate pipelines to reduce hallucinations in models like ChatGPT.

Large language models seem set to be increasingly used for information seeking tasks. If such tasks drive important health related decisions (such as what treatment to take) then it is paramount we better understand these models. This understanding includes the obvious aspect of answer correctness, but also how different prompting can sway the model. An understanding of such aspects both helps mitigate potential harms and helps inform the design of better models.

Prompts, raw result files and manual analysis are made publicly available at: `https://github.com/ielab/drchatgpt-health_prompting`.

## Limitations

Our study has a number of limitations earmarked for future work. We did not analyse the characteristics of the evidence inserted in prompts that triggered divergent answers, despite having identical stance, nor did we study the effect of different prompt formats (including the extraction of key passages from the evidence documents used) — aspects that are known to lead to variability in effectiveness in prompt learning (Zhao et al., 2021).

We did not perform fine-tuning of the LLM to specialise it to answering health questions from consumers. While we acknowledge that providers offering such a service to health consumers may investigate fine-tuning their LLMs, we instead wanted to evaluate the common case of people directly posing their health questions to ChatGPT, which is currently a realistic scenario.

In ChatGPT, like in other LLMs, answer generation is stochastic. We could have controlled for this by setting the temperature parameter of the model to zero, thus enforcing deterministic answers when the LLM was provided with the same prompt. We instead resorted to study the variability of answer generation (and the associated effectiveness) across multiple runs of the same question using the default temperature value from ChatGPT. We did this because a user of ChatGPT does not have control of the temperature parameter, and thus we wanted to accurately reflect this real scenario.

A key feature of ChatGPT is its interactivity: it supports multi-turn conversations. We, instead, only considered the single-turn setting. The ability to hold multi-turn conversations would support, for example, providing multiple evidence items, demand for a binary decision instead of an unsure position, and clarifying aspects of the answer that may be unclear.

When producing an answer, we instructed ChatGPT to also explain the reason for providing the specific advice. These explanations often included claims about research studies and medical practices; we did not validate these claims. In addition, we did not ask to attribute such claims to sources (Bohnet et al., 2022; Menick et al., 2022; Rashkin et al., 2021): correct attribution appears to be a critical requirement to improve the quality, interpretability and trustworthiness of these methods.

Finally, in our analysis we revealed that users may be affected by anchoring bias (Azzopardi, 2021) when examine ChatGPT's answers, especially those for which the model declares itself Unsure. We acknowledge that user studies should be instructed to further understand this aspect.

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

## A  Sample Prompts for RQ2

Figure 9 contains two actual prompts we used, one with support and one with contrary evidence.

**Supporting evidence**

```
Can folic acid help improve cognition
and treat dementia?

A web search for this question has
returned the following evidence, which I
provide to you in quotes:

"An important study appeared in JAMA
Psychiatry in June 2016, providing
additional evidence that high blood
levels of vitamin B12 can slow the
shrinking of the brain that commonly
occurs after age 60..."

You MUST answer to my question with one
of the following options ONLY: <Yes>,
<No>. Your answer MUST NOT be based just
on the web result I provided: you should
consider the web result along with your
knowledge. Please also provide an
explanation for your answer.
```

**Contrary evidence**

```
Can folic acid help improve cognition
and treat dementia?

A web search for this question has
returned the following evidence, which I
provide to you in quotes:

"No evidence that folic acid with or
without vitamin B12 improves cognitive
function of unselected elderly people
with or without dementia. Long-term
supplementation may benefit cognitive
function of healthy older people with
high homocysteine levels..."

You MUST answer to my question with one
of the following options ONLY: <Yes>,
<No>. Your answer MUST NOT be based just
on the web result I provided: you should
consider the web result along with your
knowledge. Please also provide an
explanation for your answer.
```

Figure 9: Evidence-biased prompts submitted to ChatGPT. Each contains the question and then evidence as a document from a web search. The document either supports the question or is contrary to the questions. Questions, documents and ground truth are from the TREC Misinformation Track.