# OpenReview forum: "Dr ChatGPT tell me what I want to hear: How different prompts impact health answer correctness"
_EMNLP/2023/Conference — EMNLP 2023 Main_

### Official Review · Reviewer_96ee · 2023-08-01

**Soundness:** 4

**Excitement:**

4: Strong: This paper deepens the understanding of some phenomenon or lowers the barriers to an existing research direction.

**Paper Topic And Main Contributions:**

This paper presents an analysis of ChatGPT performance on health-related general-public questions with and without evidence. The authors use the TREC Misinformation dataset of treatment-related questions with provided evidences, both supportive and contrary, to evaluate ChatGPT capacity of responding these questions correctly.

This paper is a first research analysis of using LLM for health answers by general public. Since ChatGPT became popular and widely used and the topic of health is sensitive and risky, it is important to evaluate how ChatGPT can perform and argument its answers. The work contains detailed analysis of different scenarios and limitations of the current model.

**Questions For The Authors:**

Question A: line 461: “a couple of occasions” how many to be precise?

Question B: footnote3: are there any statistics?

Question C: Were there quotes inside the evidences?

Question D: lines 378-408: You mention that ChatGPT is very sensitive to question phrasing with negation. Could it be possible that the negation in the prompt itself (MUST/MUST NOT) contributes to the final performance as well?

**Reasons To Accept:**

- Authors raises important topics of ChatGPT behaviour: higher sensitivity to question phrasing rather than actual keywords authored, negative influence of provided evidences for question answering.
- Well argumented decisions, detailed procedures, multiple experiments
- Important notion of preconceived bias of the user about unsure answer

**Reasons To Reject:**

- Lack of in-depth analysis of influence of evidence to ChatGPT response
- Lack of statistics and analysis of evidences

**Reproducibility:**

5: Could easily reproduce the results.

**Reviewer Confidence:**

4: Quite sure. I tried to check the important points carefully. It's unlikely, though conceivable, that I missed something that should affect my ratings.

---

> ### Author Rebuttal · Authors · 2023-08-24
>
> Thank you for the clear and specific questions. We will include the relevant statistics and comments we provide in the answers as part of a new version of the paper.
>
> ### Reviewer Question: “Question A: line 461: “a couple of occasions” how many to be precise?”
> - This was on three occasions.
>
> ### Reviewer Question: “Question B: footnote3: are there any statistics?”
> - All topics had at least 3 supporting evidence passages.
> - 23 topics had at least 3 contrary evidence passages.
> - 2 topics had only 2 contrary evidence passages.
> - 6 topics had only 1 contrary evidence passage.
> - 4 topics had 0 contrary evidence passages.
>
> ### Reviewer Question: “Question C: Were there quotes inside the evidences?”
> - Out of 184 evidence passages, 77 contain quotes (42%).
> - We understand the reason for asking this question: the evidence passed in the prompt is encapsulated in quotes, and quotes within the evidence may impact this format. Our anecdotal experience with ChatGPT is that this has little to no influence, but we did not systematically investigate this.
> - While we could have intentionally removed quotes from passages, this might create a somewhat artificial setting, in that real users may very well paste into the prompt evidence that contains quotes. Our intention was to simulate as closely as possible how real users may use and interact with ChatGPT.
>
> ### Reviewer Question: “Question D: lines 378-408: You mention that ChatGPT is very sensitive to question phrasing with negation. Could it be possible that the negation in the prompt itself (MUST/MUST NOT) contributes to the final performance as well?”
> - We apologise for what seems to us a misunderstanding. We did not change the text in the prompt that refers to the part “You MUST answer to my question”: that stayed the same.
> - We instead only changed the question text (marked in the prompt of Figures 1 and 5 as [question_text]). So, if the question in the TREC dataset was “Does inhaling steam help treat common cold?” became “Inhaling steam does not help treat common cold” . We will include all these in the Appendix.

---

### Official Review · Reviewer_XcW1 · 2023-08-04

**Soundness:** 3

**Excitement:**

2: Mediocre: This paper makes marginal contributions (vs non-contemporaneous work), so I would rather not see it in the conference.

**Paper Topic And Main Contributions:**

The research paper discuss the impact of different prompts on the behavior of ChatGPT, particularly when used for health information seeking. The paper explores how the model's responses can be influenced by the way a question is asked, as well as the type of question, and how it can potentially provide incorrect answers in crucial domains like health.

The researchers examine two main scenarios:

"Question-only": Where the model is asked a health question with no further context provided, and
"Evidence-biased": Where the model is asked a health question after being provided with additional information (or evidence) obtained from a web search result.

**Questions For The Authors:**

Lack of robust validation could lead to overestimation of the proposed method's effectiveness. This includes insufficient testing, not using appropriate datasets or baselines for comparison, or cherry-picking results. The prompting approach is unclear and required detailed explanation with architecture.

**Reasons To Accept:**

One of the main strengths of any research paper is its level of innovation. If the paper presents new methods, techniques, or concepts that advance the field of NLP, it would be of great interest to the community.  If the research can be directly applied to real-world problems, this is also a considerable strength. This work shall support safety concerns in healthcare sector with the use of prompting, the need of hour.

**Reasons To Reject:**

Lack of robust validation could lead to overestimation of the proposed method's effectiveness. This includes insufficient testing, not using appropriate datasets or baselines for comparison, or cherry-picking results. The prompting approach is unclear and required detailed explanation with architecture.

**Reproducibility:**

4: Could mostly reproduce the results, but there may be some variation because of sample variance or minor variations in their interpretation of the protocol or method.

**Reviewer Confidence:**

4: Quite sure. I tried to check the important points carefully. It's unlikely, though conceivable, that I missed something that should affect my ratings.

---

> ### Author Rebuttal · Authors · 2023-08-24
>
> Thank you for taking the time to review our paper.
>
> ### Reviewer Remark: “Lack of robust validation could lead to overestimation of the proposed method's effectiveness. This includes insufficient testing, not using appropriate datasets or baselines for comparison, or cherry-picking results.”
>
> We are confused about this remark: to what extent was our testing insufficient? Our datasets were not appropriate? Which baseline should we have included in such a study? What results have been cherry-picked?
> - We believe our dataset is appropriate: it is based on a well developed public TREC (a NIST-lead, well developed international evaluation initiative) dataset on Health Misinformation. The dataset is based on high quality medical evidence (Cochrane systematic reviews)
> - Regarding baselines: other similar LLM services that have now become available including Google Bard and Bing Chat (based on ChatGPT), but they were not yet publicly available at the time of the study (and still not via API access). Yet, these are not baselines, but alternative systems. Our goal was not to benchmark a system A vs. a system B. Instead, it was to understand and quantity answers of the most widely used LLM, ChatGPT, that people are using for health advice.
> - Regarding cherry-picking. We are unsure to what result this comment is referring. We did highlight some specific cases (“Will drinking vinegar dissolve a stuck fish bone?”), but these were to exemplify general behaviour of the model.
>
> ### Reviewer Remark: “The prompting approach is unclear and requires detailed explanation with architecture.”
> - The prompts used in our experiments are given in Figures 1 and 5; two sample prompts are provided in Figure 9 in the Appendix.
> - Based on the reviewer’s comment, we will further complement these by including a workflow/architecture diagram in the Appendix.

---

### Official Review · Reviewer_o8M9 · 2023-08-12

**Typos Grammar Style And Presentation Improvements:** Minor Typos and punctuation errors.
**Soundness:** 4

**Excitement:**

4: Strong: This paper deepens the understanding of some phenomenon or lowers the barriers to an existing research direction.

**Missing References:**

NA

**Paper Topic And Main Contributions:**

The manuscript “Dr. ChatGPT tell me what I want to hear: How different prompts impact Health answer correctness” investigates the impact of different prompts on the behavior of ChatGPT based on the TREC Misinformation dataset. The manuscript empirically investigates the bias induced in the model applied to the healthcare domain.

**Questions For The Authors:**

How is the evaluation done when considering ChatGPT effectiveness based on Yes vs. No type questions?

**Reasons To Accept:**

A. The work is valuable because it provides a deeper understanding of current state-of-the-art LLM functioning with respect to the domain (in this case, healthcare).
B. The empirical and detailed evaluation through two research Questions which are :
RQ1 – Question-only Effectiveness of ChatGPT
RQ2 – Evidence-biased Effectiveness of ChaptGPT

**Reasons To Reject:**

The limitations, such as multiple talk turns, and the inspection of the relevance/authenticity of the supportive document for prompting, could have been reasons for concern. But, since the authors have mentioned it in future research directions, I do not see any strong reason for the manuscript to be rejected.

**Reproducibility:**

5: Could easily reproduce the results.

**Reviewer Confidence:**

5: Positive that my evaluation is correct. I read the paper very carefully and I am very familiar with related work.

---

> ### Author Rebuttal · Authors · 2023-08-24
>
> Thank you for taking the time to review our paper.
>
> ### Reviewer Question: “How is the evaluation done when considering ChatGPT effectiveness based on Yes vs. No type questions?”
>
> In the experiments where ChatGPT was instructed to only produce a Yes/No answer, ChatGPT did successfully constrain the response to start with these tokens. Therefore, the response could be automatically assigned to the Yes/No classes.
>
> In the experiments where ChatGPT was instructed to produce a Yes/No/Unsure answer, we adapted the following process:
> - If the ChatGPT response started with the terms “Yes”, “No”, or “Unsure” then these were recorded automatically. (Most cases fit this condition.)
> - For the few cases where the response did not start with the terms “Yes'', “No” or “Unsure”, we manually assessed these responses.
> - The manual assessment was done by two annotators (authors). While there were no disagreements, annotators flagged some cases for joint discussion and adjudication.

---

### Meta-Review · Area_Chair_86UR · 2023-09-19

**Recommendation:** 4

**Metareview:**

The paper investigates the impact of different prompts on the behavior of ChatGPT based on the TREC Misinformation dataset. They report the bias induced in the model applied to the healthcare domain.

All reviewers agreed that the paper studies an important topic, and "support safety concerns in healthcare sector". Reviewers asked some specific questions about the study method and the statistical significance of the results, which I believe the authors sufficiently addressed in their rebuttals. Note that one reviewer who rated the paper most negatively did not update their score, but did comment that "I think the paper is good to go now."

I recommend accepting the paper into the Main track, and encourage the authors to add a figure illustrating the workflow, extending the related work session, and providing clarification to the human eval.

---

### Decision · Program_Chairs · 2023-10-07

**Decision:**

Accept-Main

**Comment:**

The paper investigates the impact of different prompts on the behavior of ChatGPT based on the TREC Misinformation dataset. They report the bias induced in the model applied to the healthcare domain.

All reviewers agreed that the paper studies an important topic, and "support safety concerns in healthcare sector". Reviewers asked some specific questions about the study method and the statistical significance of the results, which I believe the authors sufficiently addressed in their rebuttals. Note that one reviewer who rated the paper most negatively did not update their score, but did comment that "I think the paper is good to go now."

I recommend accepting the paper into the Main track, and encourage the authors to add a figure illustrating the workflow, extending the related work session, and providing clarification to the human eval.